# MAP the Blockchain World: A Trustless and Scalable Blockchain Interoperability Protocol for Cross-chain Applications

## Abstract

Blockchain interoperability protocols enable cross-chain asset transfers or data retrievals between isolated chains, which are considered as the core infrastructure for Web 3.0 applications such as decentralized finance protocols. However, existing protocols either face severe scalability issues due to high on-chain and off-chain costs, or suffer from trust concerns because of centralized designs.

In this paper, we propose MAP, a trustless blockchain interoperability protocol that relays cross-chain transactions across heterogeneous chains with high scalability. First, within MAP, we develop a novel *cross-chain relay* technique, which integrates a unified relay chain architecture and on-chain light clients of different source chains, allowing the retrieval and verification of diverse cross-chain transactions. Furthermore, we reduce cross-chain verification costs by incorporating an optimized zk-based light client scheme that adaptively decouples signature verification overheads from inefficient smart contract execution and offloads them to off-chain provers. For experiments, we conducted the first large-scale evaluation on existing interoperability protocols. With MAP, the required number of on-chain light clients is reduced from $O(N^2)$ to $O(N)$, with around 35% reduction in on-chain costs and 25% reduction for off-chain costs when verifying cross-chain transactions.

To demonstrate the effectiveness, we deployed MAP in the real world. By 2024, we have supported over six popular public chains, 50 cross-chain applications and relayed over 200K cross-chain transactions worth over 640 million USD. Based on rich practical experiences, we constructed the first real-world cross-chain dataset to further advance blockchain interoperability research.

## Keywords

Web 3.0, Blockchain, Interoperability, Cross-chain Applications

## 1 Introduction

Blockchain is a decentralized ledger technology that uses cryptographic techniques and consensus mechanisms to achieve Byzantine Fault Tolerance (BFT), enabling decentralized trust and secure data sharing. Leveraging the philosophy of blockchain, the next generation of the web, known as Web 3.0, is being built. In recent years, a wide range of Web 3.0 applications are emerging, including cryptocurrencies, which revolutionize digital money, Decentralized Finance (DeFi) protocols that disrupt traditional financial systems, immersive virtual environments in the Metaverse, and various decentralized applications (DApps) [17] [16] [20].

**The Problem**. With the rapid development of Web 3.0, on-chain data and assets are increasingly being distributed across multiple blockchains. According to statistics, there are already over 1,000 public blockchains in the market, hosting more than 10,000 types of on-chain assets [41]. This extensive distribution creates a critical need for blockchain interoperability protocols, which enable the retrieval and transfer of on-chain data and assets between source

and destination chains through cross-chain transactions [33] [40]. With interoperability, conventional DApps could leverage data and assets from multiple chains simultaneously, thereby supporting a wider range of applications. For example, cross-chain DeFi services can increase liquidity and offer diversified financial services by integrating assets from different chains, such as Non-Fungible Tokens (NFTs), cryptocurrencies, and real-world assets (RWAs). These assets can be exchanged in a unified manner [42]. Additionally, an interoperable Metaverse could enable users to access various virtual worlds, enriching their experiences across different platforms [22].

There are three major challenges when making chains interoperable: *trust requirement*, *expensive verification*, and *chain heterogeneity*.

**Trust Requirement**. When processing cross-chain transactions, the interoperability protocol must maintain the same level of BFT security as typical public blockchains to avoid compromising overall security. This implies that the protocol should be decentralized and trustless. However, achieving this level of security is challenging, as the protocol must handle complex tasks such as cross-chain transaction retrieval, processing, and verification, while maintaining consistency and liveness. As a result, many solutions are centralized or semi-centralized, such as notary schemes and committee-based protocols [28] [35]. These are widely used by crypto exchanges but are vulnerable to internal corruption and attacks due to their reliance on trust. For example, one of the largest multi-party computation (MPC)-based cross-chain bridges, Multichain, was severely exploited, leading to a loss of over 120 million USD, allegedly due to compromised keys within its committee [38][37][49].

**Expensive Verification**. As different blockchains do not trust each other, they must verify every incoming cross-chain transaction to ensure the transaction is valid and confirmed on the source chain. However, this verification process can be expensive and inefficient, particularly when it is performed on-chain, as it involves numerous complex cryptographic operations and the storage of block headers. For example, verifying an Ethereum Virtual Machine (EVM)-compatible transaction through an on-chain Light Client (LC) consumes approximately 18 million gas, which is equivalent to about 60 USD on Ethereum at the time of writing [19]. This high cost is mainly due to the storage of public keys and the signature verification process. Although cutting-edge solutions aim to reduce on-chain costs by zk-SNARKs, they still require significant off-chain computational resources for proof generation [19] [46] [43].

**Chain Heterogeneity**. Connecting heterogeneous chains via interoperability protocols presents additional challenges. Heterogeneous chains differ in their underlying components, such as smart contract engines, supported cryptographic primitives, parameters, and transaction formats. As a result, they cannot directly verify and confirm transactions from one another. For instance, an EVM chain like Ethereum cannot directly verify transactions from Solana because the EVM lacks support for the multi-signature scheme

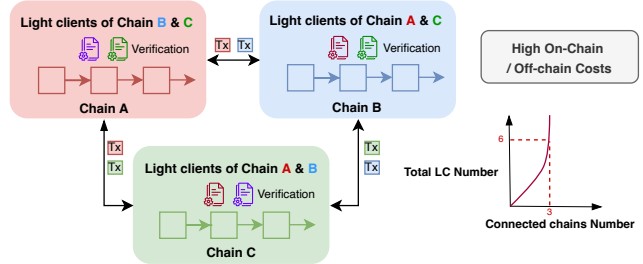

Figure 1. To connect three chains A, B, and C, LC-based protocols must deploy the LCs of chains B and C on chain A to allow it to verify transactions from those chains (and same for chains B and C), resulting in total 3*2=6 LCs needed ($O(N^2)$). Besides, it also poses heavy on-chain or off-chain costs when verifying transactions.

used in Solana transactions. Therefore, existing solutions either only support specific chain types [44] [18], or require significant modifications on the underlying components of chains to achieve compatibility [24], which are both not feasible for in-production public chains. LC-based bridges may suffer less from compatibility issues, but still need to redundantlh deploy LC contracts on each chain [19] [48] [46], as shown in Figure1. This approach incurs quadratic complexity $O(N^2)$ when extending to additional chains, thus posing huge gas consumption and development burdens.

**Our Approach**. In this paper, we introduce MAP, a scalable and trustless blockchain interoperability protocol. At a high level, MAP aims to minimize the computational costs when scaling to new chains while maintaining decentralized security, without any underlying modifications on chains. Specifically, MAP designs a novel relay chain architecture as the intermediary to relay cross-chain transactions from source chains to destination chains. By this, connecting heterogeneous chains only need to deploy their on-chain light clients. To reduce the on-chain and off-chain costs when verifying transactions, we propose an optimized zk-based light client scheme, *hybrid light client*, which adaptively decouples the workloads of Boneh-Lynn-Shacham (BLS) signature and proof verification based on their diverse performance in on-chain smart contracts and off-chain circuits.

**Contributions**. In summary, MAP makes the following contributions:

- MAP introduces a unified relay chain to facilitate cross-chain transactions between heterogeneous chains, achieving decentralized security while reducing the required number of on-chain LCs from $O(N^2)$ to $O(N)$. Furthermore, the relay chain renders MAP chain-agnostic. When extending to new chains, only corresponding on-chain light clients are required to deploy.
- We develop a hybrid light client scheme based on zk-SNARKs that reduces both the on-chain and off-chain costs of verifying cross-chain transactions. We adaptively decouple the verification workloads of BLS signatures and proof generation based their performance in on-chain smart contracts and off-chain circuits. This scheme achieves a reduction in on-chain costs by 35% and off-chain costs by 25% compared to the existing state-of-the-art works.

- We evaluate the performance and security of MAP. Specifically, for performance, we are the first to perform large-scale measurements on existing interoperability protocols. For security, besides the cross-chain liveness and consistency proof, we identify and discuss a new security issue named *inter-chain security degradation* between interoperable chains.
- We deployed MAP on six public chains and support over 50 cross-chain applications, relaying over 200K real-world cross-chain transactions, worth over 640 million USD. Base on such practical experiences, we construct the first cross-chain dataset, *BlockMAP*[1], containing over 150k cross-chain transactions across six chains. We also open-sourced all the codes of MAP (over one million lines), accompanied by detailed documentations[2].

## 2 Related Works

**Centralized/Committee-based Protocols**. To enable efficient interoperability, centralized designs are widely adopted by native protocols. Notary schemes directly host clients' tokens in custodial wallets and designate an authority (such as crypto exchanges) to facilitate their exchange efficiently [3] [9]. Similarly, committee-based protocols, such as MPC bridges and vote-oracle bridges[28] [35], appoint a small group of off-chain committees to verify and vote on cross-chain transactions, offering more decentralized features compared to notary schemes. Despite their convenience and efficiency, both solutions rely on trusting off-chain entities, which are usually not transparent and permissioned, making them vulnerable to internal corruption and attacks [28].

**Chain-based Protocols**. To further reduce the needed trust, chain-based protocols are developed, which feature at processing cross-chain transactions fully on-chain, thus making the protocols trustless. However, these protocols typically suffer from expensive verification and chain heterogeneity. Hash-Time Lock Contracts (HTLCs) are pioneering peer-to-peer protocols that allow users to deploy paired contracts on two chains to control asset release. However, HTLCs lack efficiency [2] because they require manual peer matching, enforcing users to wait for another user with the same token swap demand. As a result, HTLCs are rarly used to support large-scale cross-chain applications. Polkadot and Cosmos (Blockchain of Blockchain, BoB) employ hubs to process cross-chain transactions efficiently [44] [18], but these hubs only support their own specific homogeneous chains. HyperService [24] proposes a cross-chain programming framework, but it still requires significant modifications to the underlying components of heterogeneous chains, which is not feasible for in-production chains. LC-based bridges [19] are currently the mainstream protocols that deploy light clients (LCs) on each chain to verify cross-chain transactions. However, the internal verification workload of on-chain LCs is extremely expensive. Zero-Knowledge (ZK)LC-based bridges [46] [43] attempt to reduce on-chain costs by moving verification to off-chain provers using zk-SNARKs. Unfortunately, this requires intensive computing power and multiple distributed servers due to the large circuit size of signature verification. Additionally, all LC-based protocols face high scaling costs due to redundant LCs.

---

[1]https://zenodo.org/records/13928962
[2]https://github.com/mapprotocol

**Cross-Sharding**. Another related line of work involves sharding techniques in blockchain databases [34] [31] [29] [15] [47] [21]. In these works, cross-shard processing techniques are developed to retrieve transactions from different shards. While these techniques share similarities with cross-chain transaction processing, the key distinction is that they only consider single blockchain scenarios, where all nodes trust each other and only simple transaction verification (such as Merkle proof verification) is required. In contrast, in cross-chain scenarios, blockchains that do not trust each other, and require complicated verification.

## 3 Preliminaries

**PoS-BFT Consensus** Proof of Stake with Byzantine-Fault Tolerance (PoS-BFT) consensus has become a best practice for blockchain development due to its high energy-efficiency and security in recent years. It requires nodes (validators) to deposit funds as stakes to be qualified to participate in the consensus and to guarantee security. PoS-BFT consensus procedures typically operate and iterate in epochs. At the beginning and end of each epoch, validators are rotated and elected as committees by the PoS mechanism. During the epoch, there will be a fixed period of time for the committees to validate, agree and finalize proposed blocks according to BFT algorithms and PoS mechanism[25][11].

**Light Client**. The light client serves as alternative option for resource-constrained devices such as mobile phones to run blockchain nodes. It only syncs and stores block headers to reduce storage and computation overheads. Therefore, only partial functions of full nodes are available, such as transaction query and verification, while the costly consensus and mining procedures are usually excluded[8].

**Aggregate Signature** Aggregate signature (or aggregate multi-signature) refers to the signature scheme that supports batch verification on signatures with public keys to reduce overheads[4][5]. In aggregate signature schemes, multiple signatures are aggregated as one signature, which are further verified by an aggregated public key. BLS signature and its variants currently are widely used in PoS-BFT chians due to their high efficiency.

**Zero-knowledge Proof** The Zero-Knowledge Proof (ZKP) system is a cryptographic protocol that allows a prover to prove to a verifier that a given statement is true without disclosing any additional information besides the fact that the statement is indeed true or false. ZKP systems typically need to express and compile the statement proof procedures into circuits with constraints (gates) to generate proofs, which is complex and computationally expensive [36][27] [23].

## 4 System Model and Goals

**Interoperability Model**. In MAP, we consider the most general interoperability model that exists in most cross-chain applications. In this model, there are typically two types of chains to achieve interoperability through a *relay* process: the source chain $\mathbb{SC}$ and the destination chain $\mathbb{DC}$. $\mathbb{SC}$ is the initiating entity of the relay process, which first receives and acknowledges cross-chain transactions *ctx* from users and DApps. Then, a *blockchain interoperability protocol* is deployed between $\mathbb{SC}$ and $\mathbb{DC}$, responsible for relaying *ctx* between them.

**Transaction Model**. Interoperability between blockchains is implemented in the form of cross-chain transactions *ctx* in MAP. A *ctx* is a blockchain transaction from $\mathbb{SC}$ to $\mathbb{DC}$ containing the message or asset to be transferred. Formally it is defined as $ctx = \{\mathbb{DC}, payload\}$. The $\mathbb{DC}$ field is the chain id of $\mathbb{DC}$, which identifies the destination of *ctx*. *payload* field is the actual regarding the two types of *ctx*. When a *ctx* is an asset transaction, its *payload* contains the specific asset type, the amount, and the asset operation instructions; when a *ctx* is a message transaction, its *payload* contains the smart contract calls. In MAP, different types of *ctx* are handled in the identical way.

**Design Goals**. MAP has the following design goals:

1. **Trustless**. Maintaining the same level of BFT security as typical public blockchains.
2. **Scalability**. Gas-efficient and computationally efficient when processing cross-chain transactions and scaling to new chains.
3. **Chain-agnostic**. When extending to new chains, no underlying modifications needed except deploying new smart contracts.

## 5 MAP Protocol

### 5.1 Overview

As shown in Figure 2, there are two pipelined phases of cross-chain relay in MAP:

(Phase 1. $\mathbb{SC}$ - $\mathbb{RC}$). First, cross-chain transactions *ctx* are firstly committed by users or DApps and confirmed on the source chain $\mathbb{SC}$ (❶). Then, an off-chain server *prover* will proactively monitor this confirmation event and retrieve the *ctx* with its associated proofs issued by $\mathbb{SC}$, such as headers and Merkle proofs (❷). Then the *ctx* and its proofs are sent to the unified relay chain $\mathbb{RC}$ by *prover* for generating proofs (❸).

The *unified relay chain* $\mathbb{RC}$ is an intermediary blockchain that processes cross-chain transactions between source and destination chains in a unified manner. More specifically, $\mathbb{RC}$ integrates multiple *hybrid on-chain LCs* of each $\mathbb{SC}$ (our zk-SNARKs-based optimized version of LCs, details in §5.3), which receive *ctx*s from *prover* and verify whether they are legal and already confirmed on $\mathbb{SC}$ (❹). After the verification, the *ctx*s are temporarily confirmed and appended to $\mathbb{RC}$.

(Phase 2. $\mathbb{RC}$ - $\mathbb{DC}$). Similar with phase 1, this is another off-chain server *prover* retrieving *ctx*s from $\mathbb{RC}$ (❺). *prover* generates the proofs of *ctx*s for verification on the destination chain $\mathbb{DC}$ (❻). On each $\mathbb{DC}$, an *identical* hybrid on-chain LC of $\mathbb{RC}$ is deployed, which verifies whether *ctx*s confirmed on $\mathbb{RC}$. Finally, the *ctx*s initially committed to $\mathbb{SC}$ are eventually confirmed on $\mathbb{DC}$, thus finalizing the entire cross-chain relay procedure (❼).

Note that there could be multiple $\mathbb{SC}$ and $\mathbb{DC}$ pairs in MAP, and the relay process is executed in the same way for each pair. Besides, $\mathbb{SC}$ and $\mathbb{DC}$ are relative, which means they could be switched in reversed relay processes in MAP (by deploying LCs of $\mathbb{RC}$).

### 5.2 Unified Relay Chain

**Insights**. To address the trust and heterogeneity challenges, we present on two key insights on designing the architecture of blockchain interoperability protocols: (1) Only a BFT system can

**Figure 2. Overview of MAP:** We introduce a unified relay chain as a framework to facilitate cross-chain communications, which continually retrieves and verifies cross-chain transactions from source blockchains, including their blocks, transactions, and related proofs. This procedure is executed by the normal on-chain light client and our hybrid light clients based on zk-SNARKs, which are implemented by smart contracts with off-chain provers.

maintain the same security level with connected blockchains, thus avoiding degrading overall security. Therefore, the overall architecture must be BFT-secure, such as a blockchain. (2) For decentralized protocols like (ZK)LC-based bridges, the number of LCs on each chains are actually the *overlapping* and *redundant*. That is, each chain only consider how to verify other chains from their own perspective (i.e., deploy other chains' LC linearly), which ignores that the same type of LC may be deployed for multiple times from global view. For example, as shown in Figure 1, each types of LCs are actually deployed twice. Therefore, if the architecture is able to verify transactions from different heterogeneous chains in a unified way, instead of overlapping and redundant, the heterogeneity challenge will be effectively resolved.

**Architecture**. To consolidate the above insights, we introduce the relay chain $\mathbb{RC}$ as the cross-chain intermediary in MAP. First, $\mathbb{RC}$ itself is blockchain primarily responsible for receiving transactions from the source chain, verifying them, and forwarding verified transactions to the destination chain. This relay chain fundamentally ensures that MAP maintains decentralized security and trustworthiness.

Moreover, to address the challenge of chain heterogeneity, we adopt a *unified processing* strategy that enables $\mathbb{RC}$ to efficiently verify $ctx$ from different heterogeneous chains, thus minimizing the number of LCs on $\mathbb{SC}$ and $\mathbb{DC}$. Specifically, MAP uses the on-chain LCs for cross-chain transaction verification. However, unlike existing LC-based bridges that require each of the LCs to be deployed on every other chain, we instead integrate the LCs of different chains into a single $\mathbb{RC}$. Consequently, all on-chain LCs $\Pi_{hlc}^{sc} = \langle \Pi_{hlc}^{sc_1}, \Pi_{hlc}^{sc_2}, \ldots, \Pi_{hlc}^{sc_i} \rangle$ are build on $\mathbb{RC}$ (the internal process of $\Pi_{hlc}^{sc}$ will be introduce in §5.3).

**Cross-Chain Relay**. The general process of relaying $ctx$ from source chain $\mathbb{SC}_\mathbb{I}$ to $\mathbb{DC}$ works as follows. As shown in the Algorithm 1, there are two pipelined phases.

First, for the $\mathbb{SC}_\mathbb{I} - \mathbb{RC}$ phase, after $ctx$ is committed and confirmed on $\mathbb{SC}_\mathbb{I}$, it will emit a confirmation event by outputting the block header $bh^{sc_i}$ with the Merkle tree root $r_{mkl}^{sc_i}$ (line 2). Then a *prover* between $\mathbb{SC}_\mathbb{I}$ and $\mathbb{RC}$ will monitor this confirmation event and proactively retrieve the $ctx$ and generate the proofs $\langle ctx, bh^{sc_i}, \pi_{mkl}^{sc_i}, \pi_{zk}^{sc_i} \rangle$ (line 4-5) from $\mathbb{SC}_\mathbb{I}$ and transmit them to

---

**Algorithm 1:** Unified Relay Chain in MAP

**Input:** A cross-chain transaction $ctx$ from $\mathbb{SC}_\mathbb{I}$ to $\mathbb{DC}$
**Output:** Updated $\mathbb{DC}$ by $ctx$

1 **Procedure** SourceChain($ctx$):
2   $(bh^{sc_i}, r_{mkl}^{sc_i}) \leftarrow$ confirm($ctx$, $\mathbb{SC}_\mathbb{I}$) ▷ *$ctx$ is firstly committed and confirmed on $\mathbb{SC}_\mathbb{I}$*
3   **for** *prover between $\mathbb{SC}_\mathbb{I}$ and $\mathbb{RC}$* **do**
4     retrieves $(bh^{sc_i}, r_{mkl}^{sc_i})$ emitted by $ctx$ from $\mathbb{SC}_\mathbb{I}$
5     $\pi_{mkl}^{sc_i}, \pi_{zk}^{sc_i} \leftarrow$ genProof $(bh^{sc_i}, r_{mkl}^{sc_i}, ctx)$
6     **return** transmit($ctx, bh^{sc_i}, \pi_{mkl}^{sc_i}, \pi_{zk}^{sc_i}, \mathbb{RC}$)
7 **end**

8 **Procedure** RelayChain($ctx, bh^{sc_i}, \pi_{mkl}^{sc_i}, \pi_{zk}^{sc_i}$):
9   **if** $\Pi_{hlc}^{sc_i}(ctx, bh^{sc_i}, \pi_{mkl}^{sc_i}, \pi_{zk}^{sc_i}) == True$ **then**
10     $(bh^{rc}, r_{mkl}^{rc}) \leftarrow$ confirm($ctx$,$\mathbb{RC}$) ▷ *$ctx$ is verified and confirmed on $\mathbb{RC}$ by corresponding $\mathbb{SC}_\mathbb{I}$'s light client*
11     **for** *prover between $\mathbb{RC}$ and $\mathbb{DC}$* **do**
12       retrieves $(bh^{rc}, r_{mkl}^{rc})$ emitted by $ctx$ from $\mathbb{RC}$
13       $\pi_{mkl}^{rc}, \pi_{zk}^{rc} \leftarrow$ genProof $(bh^{rc}, r_{mkl}^{rc}, ctx)$
14       **return** transmit($\widehat{ctx}, bh^{rc}, \pi_{mkl}^{rc}, \mathbb{DC}$)
15     **end**
16 **end**

17 **Procedure** DestinationChain($\widehat{ctx}, bh^{rc}, \pi_{mkl}^{rc}, \pi_{zk}^{rc}$):
18   **if** $\Pi_{hlc}^{sc}(\widehat{ctx}, bh^{rc}, \pi_{mkl}^{rc}, \pi_{zk}^{rc}) == True$ **then**
19     $(bh^{dc}, r_{mkl}^{dc}) \leftarrow$ confirm($\widehat{ctx}$, $\mathbb{DC}$) ▷ *$ctx$ is finally verified and confirmed on $\mathbb{DC}$ by $\mathbb{RC}$'s light client*
20     **return** $\mathbb{DC}$
21 **end**

---

$\mathbb{RC}$ (line 6). Then $\mathbb{RC}$ verifies these transactions against the corresponding $\Pi_{hlc}^{sc_i}$ of $\mathbb{SC}_\mathbb{I}$ built on $\mathbb{RC}$. After verification, the $ctx$ are confirmed on $\mathbb{RC}$ as *intermediary cross-chain transactions* $\widehat{ctx}$.

Then, in the second $\mathbb{RC} - \mathbb{DC}$ phase, $\widehat{ctx}$ will also emit a confirmation event to $\mathbb{RC}$ by outputting the block header $bh^{rc}$ with the Merkle tree root $r_{mkl}^{rc}$ (line 9). Then a *prover* between $\mathbb{RC}$ and $\mathbb{DC}$ will get the $\widehat{ctx}$ and generate its proofs $\langle \widehat{ctx}, bh^{rc}, \pi_{mkl}^{rc}, \pi_{zk}^{rc} \rangle$ (line 11-12). These proofs are transmitted to $\mathbb{DC}$ for further verification

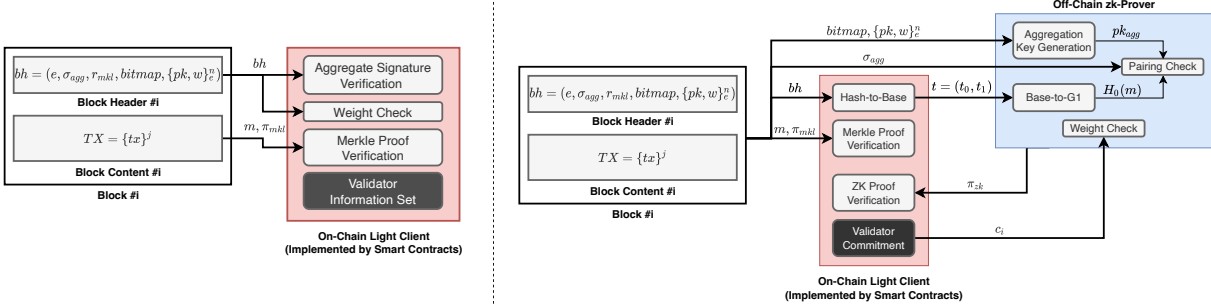

Figure 3. Our hybrid light client overperforms conventional light clients by adoptive offloading. We move the on-chain verification workloads to off-chain provers through zk-SNARKs. Meanwhile, we keep the hash operations on-chain to minimize the circuits size and proof generation time.

(line 13). The key difference here is that only one identical type of $\Pi_{hlc}^{rc}$ needs to be deployed on each $\mathbb{DC}$ (line 15) to verify $\widehat{ctx}$. This is because all $\widehat{ctx}$ are now from $\mathbb{RC}$, even though they were originally from different $\mathbb{SC}_\mathbb{I}$. After passing the verification of $\Pi_{hlc}^{rc}$, the $\widehat{ctx}$ are confirmed on $\mathbb{DC}$ as the finalized cross-chain transactions $ctx$.

**Consensus**. To ensure the decentralized security of the relay process on $\mathbb{RC}$, we make $\mathbb{RC}$ run a BFT consensus (e.g., a PoS BFT consensus like IBFT[26]). It enforces honest nodes with economic incentives (such as block rewards), while punishing malicious behavior by slashing incentives. As long as honest nodes control the majority of the total stake (e.g., greater than $\frac{2}{3}$), the $\Pi_{hlc}^{sc}$ are guaranteed to execute correctly. A detailed security analysis of $\mathbb{RC}$ is presented in §7.2.

## 5.3 Hybrid Light Client

Although introducing the relay chain can effectively reduce the required number of on-chain LCs through unified processing, the heavy on-chain LC verification workload remains a bottleneck [48] [46].

**On-chain Verification**. To explore potential optimization spaces, we analyze the costs of each procedure in normal EVM-PoS light clients. After a transaction $tx$ is committed and finalized by consensus, a block $B$ and its header $bh$ will be produced and appended on chain[11]. To prove that such $tx$ is included in $B$, the following major content needs to be inputted to normal light client $\Pi_{lc}$:

- a receipt message $m$ emitted by $tx$ inside $B$.
- a Merkle proof $\pi_{mkl}$ for $m$ extracted from $B$, which is usually provided by full nodes.
- a header $bh = (\{pk, w\}^n, \sigma_{agg}, bitmap, r_{mkl})$ that consists of:
  - an epoch number $e$.
  - a current validator information set $vs_e = \{pk, w\}_e^n$ that contains $n$ validator public keys and corresponding voting weights corresponding to $e$. When consensus entering a new epoch, a new validator information set will be updated.
  - an aggregate signature $\sigma_{agg}$ from validators signing $B$.
  - a mapping value $bitmap$ that indicates which validator actually signed $B$.

- a root hash of receipt trie $r_{mkl}$ that is computed from $m$.
- other auxiliary information such as timestamp and epoch size $E$

With above input content, the normal $\Pi_{lc}$ is defined as three algorithms (Setup, Update, Verify), as shown in Figure 3 (left):

- $vs_g \leftarrow$ Setup($para$): given system parameters, $para$, initialize $\Pi_{lc}$ in terms of the epoch size, $E$, the vote threshold, $T$, and the initial validator information, $\{pk, w\}_g^n$. Then output a validator set, $vs_g = \{pk, w\}_g^n$, that indicates the current validator set stored in $\Pi_{lc}$.
- $vs_{e+1} \leftarrow$ Update($e, vs_e, bh$): given a header $bh$ with an epoch change , verify the aggregate signature $\sigma_{agg}$ inside $bh$ and update the current validator set $vs_e$ to a new validator set $vs_{e+1} = \{pk, w\}_{e+1}^n$.
- $\{0, 1\} \leftarrow$ Verify($vs_e, m, bh, \pi_{mkl}$): given a message, $m$, emitted from $tx$ and its header, $bh$, check whether $tx$ is successfully included in $B$ through its aggregate signature $\sigma_{agg}$, vote weights, and its Merkle proof $\pi_{mkl}$. Output $\{0, 1\}$ as the result. The incremental increase in the epoch number, $e$, is also verified during the signature verification.

**Efficiency Optimization Space**. Computation and storage are the main overheads when triggering Update and Verify. For computation, aggregate signature verification is frequently performed, which is essentially operations on elliptic curves, including hashing (i.e., the Hash-to-Curve algorithm), equation evaluations, and pairing checks[6][1][13]. These operations are inefficient in EVM due to their relatively high complexity when calculating underlying fields via curve equations. For instance, currently verifying one EVM cross-chain transaction with full BLS signatures can cost up to $1 \times 10^6$ gas[48] (approximately 30 USD on ETH). For storage, $\Pi_{lc}$ needs to store $vs_e = \{pk, w\}_e^n$ persistently and frequently read them. Since most of PoS blockchains have more than 100 validators, storing and updating these data at the end of each epoch on smart contracts requires a large amount of storage space, thus consuming a expensive gas fees. Storing one validator information set requires $0.1 \times 10^6$ gas.

**Hybrid Verification**. To estimate the high gas fee consumption, we develop a hybrid verification scheme $\Pi_{hlc}$ to reduce on-chain costs using off-chain zk-SNARKs.

First, we aim to efficiently prove the two functions `Update` and `Verify` using zk-SNARKs. We compress the validator information $vs$ into a single commitment: $vs = \text{commitment}(\{(pk_0, w_0), (pk_1, w_1), \ldots, (pk_n, w_n)\})$ to reduce the on-chain storage overhead. In this way, the validator aggregate signatures of $bh$ and the corresponding voting weights must satisfy this commitment value to pass verification. One native approach to implementing zk-SNARKs for proving is to program and compile all verification procedures into circuits, i.e., input the entire block header into the circuit along with all signature verification algorithms [46] [39]. Then deploy an off-chain prover to generate the zk-proofs based on this circuit and submits them to $\Pi_{hlc}$ for verification.

However, we observe that despite $\Pi_{hlc}$ improving efficiency by shifting on-chain workloads to off-chain provers, generating zk-proofs for verifying the entire aggregate signature instead requires substantial off-chain storage and computational resources for the prover. Specifically, in this way, the circuit size for an aggregate signature verification is extremely large due to multiple complex operations such as *Hash-to-Curve* and pairing checks (typically exceeding $2 \times 10^7$ gates in existing implementations and 100 GB[12] for eight signatures). These factors also increase the proof generation time.

To optimize the off-chain costs of generating zk-proofs, we try to decouple the aggregate signature verification process and handle it separately. Specifically, the *Hash-to-Curve* algorithm in the BLS scheme hashes the message $m$ to curve points in $\mathbb{G}$, which typically consists of two steps in practical implementations:

1. *Hash-to-Base*. Input a a message $m$ and map it to possible coordinates (base field elements) through hash functions. This returns a field element $t$.
2. *Base-to-G*. Input a field element $t$ and calculate the curve point $(x, y)$ through the curve equations.

Since *Hash-to-Base* mainly consists of multiple hash operations, it can be efficiently computed through smart contract but inefficiently compiled into circuits due to its large size. In contrast, *Base-to-G* performs arithmetic operations in the finite field through elliptic curve equations, which can be relatively briefly and efficiently expressed into circuits. In this way, we improve the off-chain efficiency of zk-SNARKs based aggregate signature verification, further speed up the entire $\Pi_{hlc}$.

With the above optimization, the $\Pi_{hlc}$ is defined as the following algorithms, as shown in Figure 3(right):

- $vs_g \leftarrow \text{Setup}(para)$: given the system parameters, $para$, initialize $\Pi_{hlc}$ with the hard-coded epoch size, $E$, the vote threshold, $T$, and the initial validator information commitment, $vs_g = C(\{pk, w\}_g^n)$. Then, output a validator set, $vs_g = \{pk, w\}_g^n$, that indicates the current validator set stored in $\Pi_{hlc}$.
- $vs_{e+1} \leftarrow \text{Update}(vs_e, h, \pi_{zk})$: given header $bh$ during an epoch change, verify the aggregate signature, $\sigma_{agg}$, of $bh$. First, compute the base field elements $t = (t_0, t_1)$ in $G_1$ by hash function $H_0(bh)$, and send $t$ to the prover. After receiving $\pi_{zk}$ that satisfied $c$, update the current validator set, $vs_e$, with the new validator set, $vs_{e+1} = C(\{pk, w\}_{e+1}^n)$.

- $\pi_{zk} \leftarrow \text{GenZK}(bitmap, vs_e, \sigma_{agg}, t, )$: given extracted $bitmap$, $vs_e = \{pk, w\}_e^n$, $\sigma_{agg}$, validator set commitment $c$ from $vs_e$ and $t$ from `Update`, run a zk-SNARKs system and generate a zk-proof, $\pi_{zk}$, for $c$.
- $\{0, 1\} \leftarrow \text{Verify}(vs_e, m, h, \pi_{mkl})$: given message $m$ emitted from $tx$ and its header $bh$, verify whether $tx$ is successfully included in $B$ through its aggregate signature, $\sigma_{agg}$, and there are sufficient weights according to the stored $vs_e$ and its Merkle proof $\pi_{mkl}$. Then output $\{0, 1\}$ as the result.

## 6 Performance Evaluation

**Experiment Setup**. We set up a Google Compute Engine machine type c2d-highcpu-32 instance (32 vCPUs with 64GB RAM, ~800 USD per month) as a prover, and a e2-medium instance as prover (1 vCPUs with 4GB RAM, ~24 USD per month). For the relay chain, the hardware configuration for validator is similar with e2-standard-4 (4 vCPUs with 16 GB RAM), and requires at least $1 \times 10^6$ MAP token as stakes (~10K USD).

**Baselines and Workloads**. Since very few works provide quantitative performance evaluation results, it is difficult to find a fair baseline [36][33][40]. To this end, we perform the first comprehensive measurement and comparison of existing blockchain interoperability protocols. As shown in Table 1, we measure five key security and scalability metrics across six representative types of protocols. Based on these results, we select the state-of-the-art (SOTA) results as baselines for comparison. We set the workloads as cross-chain transactions from *Polygon to Ethereum* for comparison, which is mostly supported by existing works. For protocols that do not support such workloads (such as Polkadot), we select their popular source-destination chain pair for evaluation. For each type of workload, we measure 100 transactions and record the average result.

### 6.1 Evaluation Results

**On-chain Costs**. For on-chain costs, we mainly refer to the LC-based bridges as baselines, because they are the most common decentralized solutions [48]. For each cross-chain transaction verification, on-chain LCs require ~$1 \times 10^6$, while MAP requires only ~0.65M at the time of writing, saving ~35%. These costs are deterministic in repeated tests on smart contracts [11] [45].

**Off-chain Costs**. For off-chain costs caused by zk-SNARKs, we refer to the standard implementation using snarkjs Groth16 to prove signature verification as the baseline [12] [46] [39]. As shown in Figure 2, for eight signatures, the circuit size of the MAP prover is ~$1.57 \times 10^7$ gates, which is reduced by ~25% compared to the aforementioned baselines ($2 \times 10^7$ gates). Correspondingly, the proof generation time is also reduced by ~25% due to its linear relationship with circuit size.

**Number of On-chain Light Clients (scaling up costs)**. According to statistics[3], a PoS-BFT EVM light client requires approximately ~100K gas per validator information storage. Assuming the number of validators is 100 for each chain, then for connecting $N$ chains, LC-based bridges need to spend $10^7 \times N(N-1)$ gas to

---

[3]https://github.com/shreshagrawal/poc-superlight-client

| Evaluation Metrics | Centralized | Committee | Chain | | | | |
|---|---|---|---|---|---|---|---|
| Solutions | Binance, CoinBase[3][9] | Multichain, Celer[28][10] | EthHTLC[7], Lighting[2] | Polkadot, Cosmos[44][18] | Horizon, LayerZero,[19][48] | zkRelay, zkBridge[43][46] | MAP |
| Type | Notary | MPC | HTLC | BoB | LC | ZKLC | ZKLC+Relay |
| Security Models | Trusted | Semi-Trusted | Trustless | Trustless | Trustless | Trustless | **Trustless** |
| On-chain Costs (gas) | N/A | $0.5 \times 10^6$ | $1.5 \times 10^6$ | $0.8 \times 10^5$ | $1 \times 10^6$ | $0.3 \times 10^6$ | $0.65 \times 10^6$ **(35% less)** |
| Off-chain Costs (gates) | N/A | N/A | N/A | N/A | N/A | $2 \times 10^7$ | $1.57 \times 10^7$ **(25% less)** |
| Latency | 1s | 310s | N/A | 13s | 227s | 153s | **210s** |
| Complexity | $O(N)$ | $O(N^2)$ | $O(N^2)$ | $O(N)$ | $O(N^2)$ | $O(N^2)$ | **O(N)** |

**Table 1. Performance Comparisons of MAP and Existing Blockchain Interoperability Protocols. (Polygon to Ethereum transaction workload)**

**Table 2. Circuit size of provers for verifying different number of validator signatures**

| Number of Sigs (Validators) | Circuit Size (gates) |
|---|---|
| 4 | $0.9 \times 10^6$ |
| 8 | $15.7 \times 10^6$ |
| 16 | $25.2 \times 10^6$ |
| 32 | $49.3 \times 10^6$ |

deploy LCs. In contrast, for MAP, it is ~100K gas fixed per LC for validator information set commitment storage (no matter how many validators), which means only $2 \times 10^5 \times N$ gas is needed.

**Cross-chain Latency** We measure the end-to-end latency of cross-chain transactions relayed in MAP, from the confirmation timestamp on source chains until the confirmation timestamp on destination chains, including transaction transmission between chains, proof generation, and on-chain LC verification. As shown in Table 4, the results indicate that MAP's cross-chain latency is ~210 seconds. Compared to existing works, these results suggest that despite introducing provers and relay chain will increase latency, the on-chain LCs execution are simplified to reduce the overall latency.

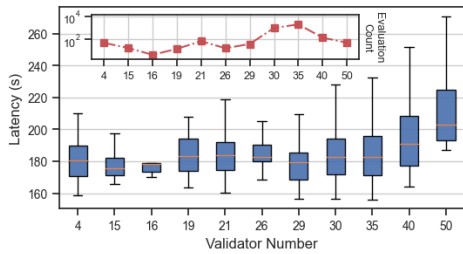

**Figure 4. Cross-chain latency under different size of validators**

**Real-world Cross-chain Dataset**. Based on the experiments and our deployment statistics, we prune and provide the first public, real-world blockchain interoperability dataset, BlockMAP[4], which consists of 150k cross-chain transactions from six popular public chains. The dataset includes several essential attributes, such as transaction direction, start and end timestamps, token types, and amounts. This dataset presents practical measurement of real-world cross-chain transactions, aiming to offer new insights and understandings for future blockchain research.

[4]https://zenodo.org/records/13928962

## 7 Security Analysis

We thoroughly analyze the security of MAP. Particularly, as previous works have extensively proved the security of a transaction will be confirmed on with liveness and consistency within a single PoS-BFT chain [32][30], we focus on demonstrating the newly introduced components (i.e., provers and relay chain) in MAP will still maintain the liveness and consistency under various attacks.

### 7.1 Assumptions

MAP works under several basic and common security assumptions in blockchain communities [36].

ASSUMPTION 1. (*PoS-BFT Threshold*). For $\mathbb{RC}$, more than $\tau = \frac{2S}{3}$ of the stakes are controlled by honest validators, where $S$ is the total stakes. This group of honest validators is always live, i.e., they will confirm ctx in a timely manner.

ASSUMPTION 2. (*Secure Cryptographic Primitives*). The cryptographic primitives used in MAP, including the BLS signature, the Groth16 zk-SNARKs, and the hash functions, are secure against probabilistic polynomial-time (PPT) adversaries. That is, no PPT adversary can generate incorrect proofs or signatures that would be accepted.

ASSUMPTION 3. (*Minimal Prover and Reachable Communication*). At least one prover is available and honest in MAP, i.e., they will correctly generate the proofs $\pi_{mkl}$ and $\pi_{zk}$ and transmit cross-chain transactions ctx between chains, i.e., $\mathbb{SC}$, $\mathbb{RC}$, and $\mathbb{DC}$. Additionally, we assume that the communication channels between the prover and the chains are reachable (i.e., no network partitions, though they may be insecure).

### 7.2 Liveness and Consistency

THEOREM 1. (*Cross-chain Liveness*). If a valid ctx is committed to and confirmed on $\mathbb{SC}$, then it will eventually be confirmed on $\mathbb{DC}$ via MAP, assuming the above assumptions hold.

PROOF. Given a committed ctx from $\mathbb{SC}$, there are two potential cases that could prevent it from being confirmed on $\mathbb{DC}$: *Case 1*: A faulty or compromised $\overline{prover}$ refuses to generate proofs and transmit ctx between SC-RC or RC-DC. *Case 2*: Sufficient validators of RC are corrupted to force $\mathbb{RC}$ to withhold ctx, preventing it from being sent to $\mathbb{DC}$. For Case 1, by Assumption 3, at least one *prover* will transmit ctx to $\mathbb{RC}$ and $\mathbb{DC}$ (a single *prover* is sufficient for processing transactions from any number of chains). Therefore,

even if other *provers* are faulty or compromised (e.g., via DDoS attacks), $\mathbb{RC}$ and $\overline{\mathbb{DC}}$ can still receive and verify $ctx$ from the reliable *prover*. For Case 2, previous works have proven that any liveness attacks on PoS-BFT chains involving the refusal to verify transactions require at least $\frac{1S}{3}$ stakes [32][30], which is prevented by Assumption 1. Even in the case of DDoS attacks on some of the $\mathbb{RC}$ validators, since the honest validators are live and control over $\frac{2S}{3}$, they will always confirm the $ctx$ in time. As a result, $\Pi_{hlc}$ run by the validators will eventually verify $ctx$ and confirm it on both $\mathbb{RC}$ and $\mathbb{DC}$, thereby guaranteeing the overall cross-chain liveness. □

THEOREM 2. *(**Cross-chain Consistency**). If a valid ctx is committed and confirmed on* $\mathbb{SC}$ *and a* $\underline{ctx}$ *is finally confirmed on* $\mathbb{DC}$ *via* MAP, *then* $ctx = \underline{ctx}$, *assuming the above assumptions hold.*

PROOF. Given a $ctx$ from $\mathbb{SC}$, there are two potential cases for consistency attacks: *Case 1*: A malicious $\underline{prover}$ generates a tampered $\underline{ctx}$ with its proofs and tries to get them accepted by $\mathbb{RC}$. *Case 2*: Adversaries directly corrupt $\mathbb{RC}$ to force it to accept a tampered $\underline{ctx}$. For Case 1, in order to pass $\Pi_{hlc}$ verification, the malicious $\underline{prover}$ would need to forge block headers (including the corresponding signatures and Merkle proofs) to generate incorrect zk-proofs. However, by Assumption 2, this is highly unlikely to succeed. Therefore, $\Pi_{hlc}$ will not accept $\underline{ctx}$ as a valid cross-chain transaction on $\mathbb{RC}$. For Case 2, corrupting $\mathbb{RC}$ to accept a tampered $\underline{ctx}$ requires controlling at least $\frac{2S}{3}$ of the validators, which is prevented by Assumption 1. Therefore, any tampered $\underline{ctx}$ will not be accepted on $\mathbb{RC}$, thus ensuring cross-chain consistency. □

### 7.3 Inter-Chain Security

Despite the analysis in §7.2 proving that cross-chain transaction verification is secure under Assumptions 1, 2, and 3, it does not fully match cross-chain scenarios. Specifically, within a single chain, the profit-from-corruption can hardly be higher than cost-to-corruption because they are calculated by the relative token value. That is, within a chain A with security threshold $\tau_A = \frac{2S_A}{3}$, it is unlikely to see a transaction with value over $\tau_A$.

We identify a new potential security issue when connecting multiple chains with interoperability protocols that may converse the above situation, which also applies to chain-based protocols but never discussed before. We name this issue *Inter-Chain Security Degradation*. We argue that the overall security of interoperable multi-chain networks is as strong as the least secure chain. For example, given three interoperable PoS-BFT chains A, B, and C, with their BFT security boundaries as $\tau_A = \frac{2S_A}{3}$, $\tau_B = \frac{2S_B}{3}$, and $\tau_C = \frac{2S_C}{3}$, the security of the entire network is $\min(\tau_A, \tau_B, \tau_C)$. This can be justified by considering the following situation: assume $\tau_B = \min(\tau_A, \tau_B, \tau_C)$. If a $ctx$ from chain A to chain B has a extremely large value $V_{extreme} > \tau_B$, the validators of chain B will be motivated to manipulate $ctx_{extreme}$ (such as double-spending), even if they were honest before (Assumption 1) and run the risk of being slashed by all the staked. Because their profit-from-corruption is now explicitly higher than cost-to-corruption. In other words, the security of chains A, B, and C is *degraded* to $V_{extreme} < \tau_B$ due to interoperability.

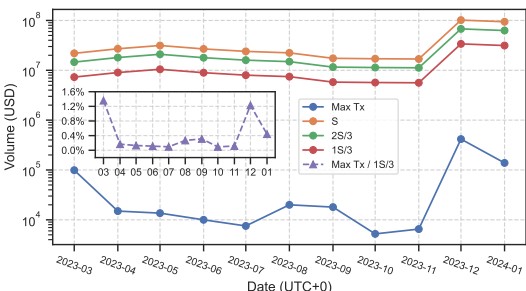

**Figure 5. Historical statistics of** MAP: **The maximum value of any single cross-chain transaction is significantly smaller than the security boundary of the relay chain**

**Discussion** Regarding MAP, this degradation requires the security of the relay chain to be strong enough (high staked value) to support cross-chain transactions. To examine this, we provide real-world statistics in MAP. As shown in Figure 5, the most valuable cross-chain transaction was a 100K USDC transfer from NEAR in March 2023[5], worth 1.3% of the total MAP stakes (7M USD). This also means MAP could still support transactions worth up to 4.67M USD. In summary, although inter-chain security degradation exists due to the interoperability, MAP's relay chain design is still highly reliable and secure in practice.

## 8 Supported Chains and Cross-chain Applications

MAP supports six major public chains: including EVM chains such as Ethereum, BNB chains, Polygon, and Conflux, and Non-EVM chains such as Klaytn and Near. By 2024, there are over 640M USD assets relayed by over 5M cross-chain transactions with MAP[6]. Over 50 industrial cross-chain applications and layer-2 projects are built[7]. Representative cross-chain applications range from cross-chain swap (Butterswap), crypto payment (AlchemyPay), liquidity aggregation (Openliq), DePINs (ConsensusCore), DeFi solutions development (Unify) [8].

## 9 Conclusion

This paper introduces MAP, a trustless and scalable blockchain interoperability protocol with practical implementations. MAP strikes a balance between trustlessness and scalability by introducing a unified relay chain architecture and optimized zk-based hybrid light clients (LCs). We conducted extensive experiments to comprehensively evaluate its performance and analyze its security. Additionally, we have open-sourced the entire MAP codebase and released the first cross-chain transaction dataset, BlockMAP. We envision MAP as a practical solution for interoperable data and networking infrastructure in the Web 3.0 era.

---

[5]https://maposcan.io/cross-chains/565
[6]https://www.maposcan.io
[7]A full list at https://www.mapprotocol.io/en/ecosystem
[8]https://www.butterswap.io/swap, https://alchemypay.org, https://www.consensuscore.com,https://openliq.com, https://unifiprotocol.com

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

## A  Implementations details

For the relay chain, we develop a client software of the unified relay chain node which is compatible to both EVM chains and Non-EVM

chains (over $1 \times 10^6$ lines of Go code[9]). To overcome the heterogeneity, we integrate the most commonly adopted cryptographic primitives and parameters in existing chains into our smart contract engine. Specifically, supported hashing algorithms include SHA-3, SHA-256, keccak256, and blake2b, while signature algorithms (or elliptic curves) include ed25519, secp256k1, sr25519, and BN256, which covers most public chains. We adopt IBFT in the relay chain, which is also well tested and widely adapted in many chains. With IBFT, we issue the token $MAPO on the relay chain, which is used to pay for the gas fees of cross-chain transactions and the block rewards for validators.

We also implement our proposed hybrid LCs together with normal LCs (six clients for six chains, totally over 180K lines of Solidity code[10]), spawning multiple smart contracts.

For off-chain provers, we use Groth16[14] to express the BLS signature verification (except *Hash-to-Base*) through Circom, alongside with our optimizations to reduce the size of the circuit[11]. First, we make BLS public keys in $\mathbb{G}_2$, while the signatures are in $\mathbb{G}_1$ to reduce the signature size. Second, as mentioned before, we move two *Hash-to-Base* functions outside of the circuit to simplify the constraints in the circuit.

## B  MAP Omnichain Service

**Motivations**. Despite MAP enables trusless and scalable cross-chain transaction relaying, it is still inconvenient and costly for developers to directly integrate MAP into DApps directly in practice. That is, directly interacting with the underlying relay chain and the on-chain light clients will require intensive domain knowledge and complicate the business logics. Moreover, as the gas fees for cross-chain transactions are not negligible, a designing pricing model for cross-chain transactions is necessary.

To this end, inspared by the role of traditional DBMS in database field, we design a middleware layer named MAP Omnichain Service (MOS) upon MAP [12]. At a high level, MOS shares some similar functionalities with DBMS for database, which aims to abstract ready-to-use services from the underlying relay chain and on-chain light clients, thus effectively manage the cross-chain transactions. There two major services provided in MOS: 1) cross-chain data management service contracts, and 2) a dynamic pre-paid pricing model.

**Service Contracts for Cross-chain Data Management**. When building DApps, it is essential to manage various cross-chain data, such as sending cross-chain transactions, addresses (senders and receivers), and inquiry emitted events (transaction states, timestamp, etc). To facilitate this, MOS provides two general service contracts as interfaces for DApps to relay cross-chain data and inquiry cross-chain data conveniently, as defined in the followings:

- `dataOut(uint256 _toChainId, bytes memory _messageData, address _feeToken)` (deployed on source chains and the relay chain)

- `dataIn(uint256 _fromChainId, bytes memory _receiptProof)` (deployed on the relay chain and destination chains)

where `_toChain` is the destination chain chain id, `_messageData` is the cross-chain data to be relayed, `_feeToken` is the address of the token type for paying the cross-chain fees. To relay data, an DApps first calls the *messageOut* on $\mathbb{SC}$ by specfiying the id of $\mathbb{RC}$, data payload, and paying the fees of the cross-chain. When the $\mathbb{SC}$-$\mathbb{RC}$ messager observes the event emitted from *messageOut*, it builds the corresponding proofs and sends them to the *messageIn* on $\mathbb{RC}$, which will future call the on-chain LCs for verification. likewise, the *messageOut* and *messageIn* will be called on $\mathbb{RC}$ and $\mathbb{DC}$, respectively, and the message data is eventually relayed.

For cross-chain data inquiry, each *messageIn* will also return the hash of converted cross-chain transactions to the DApps, which can be further used to inquery the transaction status and details on the relay chain and destination chains. For example, by inquirying, DApps can demonstrate where the cross-chain transaction is currently being processed, whether it is confirmed or not, and the final status of the transaction.

**Limited Pre-paid Pricing model**. Designing pricing model for DApps to charge cross-chain transactions is significant for business sustainability. However, one major challenge is that the gas is separately consumed on each chains and hard to be precisely estimated in advance as the token price is dynamic. To address these, in MOS, we propose a pre-paid pricing model that charges the whole cross-chain transaction fees on the relay chain and the destination chain once the cross-chain transaction is confirmed on the source chain. Specifically, the pricing model $F(amount)$ is defined as follows:

$$F(amount) = \begin{cases} f_{\mathbb{RC}} + f_{\mathbb{DC}}, & F \leq f_{\mathbb{RC}} + f_{\mathbb{DC}} \\ k \times amount, & f_{\mathbb{RC}} + f_{\mathbb{DC}} < F \leq F_{max} \\ F_{max}, & F > F_{max} \end{cases}$$

where $F$ is the total cross-chain transaction fee. In normal scenarios, $F$ is decided by the cross-chain transaction token *amount* and its percentage coefficient $k$ (in MAP we usually take 0.02-0.03). In case of the cross-chain transactions only contain negligible token amount (e.g., a transaction for calling smart contracts instead of transferring tokens), the fee is set to be the sum of the basic gas fees $f_{\mathbb{RC}} + f_{\mathbb{DC}}$ for covering transaction processing fees on the relay chain and the destination chain. Besides, for cross-chain transactions containing extreme amounts of token, we set a upper bound $F_{max}$ to avoid charging unreasonable expensive fees. In this way, the pricing model ensures to cover while provding reasonable incentives for validitors to behave honestly the relay chain.

## C  Future work

In the future, we plan to extend MAP to support Bitcoin, the most renowned and valuable cryptocurrency project, enabling Bitcoin assets to be operable outside the original network and avoiding costly transaction processing.

---

[9]https://github.com/mapprotocol/atlas

[10]https://github.com/mapprotocol/atlas, https://github.com/mapprotocol/map-contracts/tree/main/mapclients/zkLightClient, and https://github.com/zkCloak/zkMapo

[11]https://github.com/zkCloak/zkMapo

[12]https://github.com/mapprotocol/mapo-service-contracts

