# OpenReview forum: "MAP the Blockchain World: A Trustless and Scalable Blockchain Interoperability Protocol for Cross-chain Applications"
_ACM.org/TheWebConf/2025/Conference — WWW 2025 Oral_

### Official Review · Reviewer_TZrP · 2024-11-25

**Novelty:** 3
**Technical Quality:** 3

**Review:**

Short summary:
The paper presents an alternative to bridging blockchains directly using clients and instead propose using a relay blockchain. The authors present the architecture and the core relay algorithm, as well as a modified signature verification method.

Overall this paper would be more suited for the Security track. This is why I chose "2" for scope, because I believe this is the wrong track. I also believe the link to the authors' GitHub project on page 2 may violate anonymization. Finally, I am not sure whether the publicly released transaction data raises ethical issues and the authors do not discuss this at all.

The paper is sometimes difficult to read and put into context for readers without any blockchain background. What is especially unclear to me is how MAP is supposed to be different from Blockchain of Blockchain projects.

Other nits:
- Citation [12] is a dead link
- All links have an access date from Nov. 2023, which presumably means the authors did not bother looking at all the links again for this submission
- What does zkSNARK stand for?
- On page 4 it is not clear what the "I" in "SC_I" is supposed to stand for.
- Figure 2 is misleading, because you would need 6 verifiers to achieve all the interopability "directions" shown in Figure 1.
- The line numbers on page 5 do not match the ones in algorithm 1.
- Algorithm 1 implies to me that we iterate over all provers ("for loop") but I am not sure whether this is the actual intention.
- In section 7.1 the authors cite [36] which does not actually define what the authors claim it does ("basic and common security assumptions in blockchain communities") as far as I can tell
- Figure 2 on page 6 should be Table 2
- Table 4 in the text on page 7 should be Table 1
- On page 6, the authors mention a baseline of 2*10^7 gates, which I was not able to find in any of the related work cited in the section.
- The x-axis in figure 4 seems to be chosen randomly and without justification.

Strengths:
- Evaluation with real deployment
- The approach does appear to reduce the number of components from O(n^2) to O(n)
- Some transaction data is made publicly available

Weaknesses:
- The paper has claims without sources or cites sources that do not actually support the claim.
- The authors do not make clear who would run a relay chain and who would benefit from running it. Another aspect left out is how the economics of the relay chain are influenced by the chains it relays between.
- I believe that attempting to transfer assets between chains with a relay chain might require discussing chain throughput issues.
- It is not clear to me why LC-based bridges need 100 validators for each chain, which is reflected in the gas cost, but this is not required for MAP.
- The latency numbers in table 1 are suspicious (especially 1 second for centralized approaches) and it is unclear how they were measured. Similar problems with figure 4. Overall, the measurement methodology underlying the latency numbers needs to be made clear.
- Overall, the evaluation is not convincingly scientific.
- The liveness proof seems like a tautology because the authors simply define that there is always one prover available.
- The transaction model and signature verification schemes are difficult to follow without background knowledge.
- Writing needs to be improved to make many parts easier to understand..

Overall, the optimization from O(N^2) to O(N) seems to be mainly about the number of validators, while the actual savings of this approach seems to be in gas cost (i.e., money) at the cost of having to run a whole relay chain. It is not clear what the cost of running such a chain is. Specifically, the economics of MAP tokens are not discussed at all, and how their value is defined. The paper reads mainly like one of those whitepapers that seem to be common in the blockchain community, and not like a rigorous scientific analysis. At the same time, the actual cryptography part seems rather short, and the inter-chain security section seems interesting and should probably be expanded and analyzed further.

**Questions:**

1. How is MAP different from Blackchain of Blockchain approaches?
2. How did you measure the latency in table 1?
3. Why is there no obvious or straightforward relation between median latency and number of validators in figure 4?
4. How is the relay chain affected by certain blockchains having lower throughput than others?
5. What are the economics of MAP token? My superficial understanding would be that MAP is L2, and thus monetary (i.e., monetary value) aspects seem important.
6. What makes the systems track the correct track for this paper over the security track?

**Reviewer Confidence:**

2: The reviewer is willing to defend the evaluation, but it is likely that the reviewer did not understand parts of the paper

**Scope:**

2: The connection to the Web is incidental, e.g., use of Web data or API

---

### Official Review · Reviewer_QZbv · 2024-11-27

**Novelty:** 5
**Technical Quality:** 5

**Review:**

# Paper Summary
The paper proposes MAP, a novel blockchain interoperability protocol designed to facilitate cross-chain transactions while addressing challenges related to trust, scalability, and chain heterogeneity. The authors introduce a unified relay chain architecture that leverages zkSNARK-based hybrid light clients for cost-efficient cross-chain transaction verification.
The protocol reduces the number of on-chain light clients from $O(N^2)$ to $O(N)$, a theoretically significant improvement. Furthermore, the paper highlights real-world adoption and contributions, including the first large-scale dataset for blockchain interoperability research.


# Strength
+ The paper tackles a highly relevant topic and presents it in a clear and accessible manner, making it easy to follow.
+ The theoretical reduction from $O(N^2)$ to $O(N)$ on-chain LC is an important theoretical improvement.
+ The real-world deployment is impressive.

# Weakness

- The comparison with existing protocols (e.g., Table 1) raise concerns about fairness and the significance of reported performance improvements.
- The evaluation metrics are limited and not intuitive.
- The system model and architecture are not clearly described.


# Detailed comments

The paper is well-written and provides a clear narrative about the challenges and solutions. However, the comparison with other schemes (e.g., Table 1) raises several points of confusion:

1. On-chain costs (gas)
The paper compares MAP with LayerZero [48], but zkBridge [46] performs better in terms of on-chain gas costs. Why was zkBridge not used as the main point of comparison?

2. Complexity
While MAP reduces the number of on-chain light clients to $O(N)$, the real-world impact of this reduction is unclear. Section 6 suggests a relationship with on-chain gas costs, but zkBridge achieves lower on-chain gas usage despite its O(N^2)$ complexity.

3. Off-Chain Costs (gates):
MAP's reported reduction in zk-SNARK gate complexity is not intuitive. How does the number of gates translate into computational or communication costs? zkBridge's reported results, including proof size (Bytes) and communication overhead (GB), provide more concrete insights. As a good example, zkBridge provides a useful reference for such evaluations.

4. Latency
The end-to-end latency of MAP (~210 seconds) is worse than zkBridge, which raises confusion about MAP's overall performance benefits.

These points collectively make it difficult to assess the significance of MAP’s improvements over zkBridge.

### Other clarity issues:

- The assumption that at least one prover is honest (Assumption 3) seems overly optimistic in adversarial settings. But I think this is actually a clarity issue: details on the number of provers (Figure 2 and the implementation suggests two) and their roles are unclear. Referring to zkBridge’s treatment of assumptions may help improve this section.

- The system model described in Section 4 does not sufficiently integrate MAP’s architecture (which appears later in Sec. 5.2). It is very unclear what exact part relates to the MAP scheme from Sec. 4.

- The implementation details are not clear. For example, how many servers are used to construct the relay chain?

> We set up a Google Compute Engine machine type c2d-highcpu-32 instance (32 vCPUs with 64GB RAM, ~800 USD per month) as a prover, and a e2-medium instance as prover (1 vCPUs with 4GB RAM, ~24 USD per month).

- What's the difference of the two _prover_?


### Some typos and formatting issues (just to list a few)
- Citation formatting is inconsistent (e.g., Page 2, Line 195; Page 3, Lines 255, 262, 266, 277).

- Page 2, line 140, Figure1 -> Figure 1

- Page 3, Line 245: Missing a period. The format of this level of section titles is inconsistent throughout the paper

- Page 4, line 393, build -> built

**Questions:**

Please clarify MAP's improvements over zkBridge:
1. For on-chain costs, why choose LayerZero as the baseline instead of zkBridge, which has better on-chain gas performance?
2. How does reducing complexity improve real-world performance, given zkBridge achieves lower gas usage with $O(N^2)$?
3. How does the reduction in zk-SNARK gates affect proof size, computational cost, or communication overhead? Can you provide these metrics?
4. Why does MAP have higher latency (~210s) than zkBridge? Does this limit its applicability in real-time scenarios?

**Reviewer Confidence:**

2: The reviewer is willing to defend the evaluation, but it is likely that the reviewer did not understand parts of the paper

**Scope:**

4: The work is relevant to the Web and to the track, and is of broad interest to the community

---

### Official Review · Reviewer_yuXL · 2024-11-30

**Novelty:** 4
**Technical Quality:** 6

**Review:**

This work proposes MAP, a trustless blockchain interoperability protocol, which provides high scalability to cross-chain transactions. With the proposed protocol, the required number of on-chain light clients and costs for verification are significantly reduced. From a thematic perspective, this work aligns with the interests of the WWW community. From a systems perspective, the work is highly robust. As I am not an expert in this field, I am unable to provide critical feedback. Nevertheless, this manuscript is very well-written, and I have learned a lot from reading it.

### Advantages
1. Motivation and scenarios are clearly presented with appropriate illustrations.
2. The processes in the proposed protocol are summarized with a well-explained algorithm.

### Disadvantages
1. Section 8 reads like a business report, if not important, it could be omitted, otherwise, it may be incorporated into existing sections.
2. There is a lack of a table summarizing all the notations, making it difficult to identify the meanings of the current symbols.

**Questions:**

How is the number of light clients typically determined? This information is important for assessing the true significance of the complexity reduction.

**Reviewer Confidence:**

1: The reviewer's evaluation is an educated guess

**Scope:**

3: The work is somewhat relevant to the Web and to the track, and is of narrow interest to a sub-community

---

### Official Review · Reviewer_43kW · 2024-12-01

**Novelty:** 6
**Technical Quality:** 6

**Review:**

The paper identifies challenges in existing blockchain interoperability protocols, including trust issues, scalability limitations, and high costs associated with cross-chain transactions. It introduces a hybrid light client scheme MAP that utilizes zero-knowledge proofs (zk-SNARKs) to reduce both on-chain and off-chain costs of verifying cross-chain transactions. This approach allows for efficient transaction processing while maintaining security. The protocol achieves a 35% reduction in on-chain costs and a 25% reduction in off-chain costs compared to existing solutions.

Strength:
Real-world impact: MAP has been deployed on six public chains, supporting over 50 cross-chain applications and facilitating more than 200,000 real-world cross-chain transactions valued at over 640 million USD. The authors also constructed a dataset, BlockMAP, containing over 150,000 cross-chain transactions.
Security considerations: The paper discusses security aspects, including cross-chain liveness and consistency, and introduces a new security issue termed inter-chain security degradation, which arises from the interactions between interoperable chains.
Comprehensive measurement: The authors conducted a thorough evaluation of existing interoperability protocols, measuring key security and scalability metrics to establish baselines for comparison.

Comments:
The introduction of the inter-chain security degradation issue raises concerns about the overall security of the network when multiple chains are interconnected. It would be beneficial if the paper could provide comprehensive discussion about the direction of solutions or mitigations for this vulnerability, addressing potential risks.

**Questions:**

Detailed in the comments above

**Reviewer Confidence:**

2: The reviewer is willing to defend the evaluation, but it is likely that the reviewer did not understand parts of the paper

**Scope:**

3: The work is somewhat relevant to the Web and to the track, and is of narrow interest to a sub-community

---

### Official Review · Reviewer_XJ3x · 2024-12-03

**Novelty:** 6
**Technical Quality:** 7

**Review:**

This paper presents MAP, a scalable protocol for cross-chain applications that is based on a dedicated relay blockchain, i.e., connectors only have to be tailored for "moving" assets from different source blockchains to the relay blockchain, and one uniform connector can be used to forward the assets from the relay blockchain to a target blockchain. MAP relies on zk-SNARKs that are verified as much as sensible off-chain to establish trust in those bridges in a scalable manner. MAP already has an impressive real-world deployment and utilization.

The basic idea is straight-forward and sensible to reduce complexity regarding the integration of new blockchains after deployment of MAP. Further, the optimizations are sensible to reduce on-chain processing costs (i.e., gas costs).

The paper is mostly well-written, even though there are notable glitches here and there (e.g., missing or wrong words, see below for a non-comprehensive list). In the same vein, spaces are sometimes missing in front of citations and \cite{A,B,C} should be used instead of \cite{A}\cite{B}\cite{C}.

(Further) Minor comments:
- Figure 1: Math mode is missing for 3*2=6
- Section 1: "Figure1" (missing space)
- Section 1: "(over one million lines)"; I suggest dropping this not really necessary detail
- Section 2: "which feature at processing"
- Section 2: "rarly"
- Section 2: "Zero-Knowledge (ZK)LC-based"
- Section 3: Zero-knowledge proofs are a much more vast framework than one simple "cryptographic protocol"
- Section 4: "is the actual regarding"
- Section 5.2: "are actually the overlapping and redundant"
- Section 5.2: "each chain only consider how"
- Section 5.2: "each types of LCs"
- Section 5.2: "RC itself is blockchain"

**Questions:**

- Would it be possible/hard to also incorporate traditional blockchains such as Bitcoin, or is the smart-contract functionality something that cannot be worked around?
- What are the incentives to contribute to the relay blockchain?

**Reviewer Confidence:**

2: The reviewer is willing to defend the evaluation, but it is likely that the reviewer did not understand parts of the paper

**Scope:**

4: The work is relevant to the Web and to the track, and is of broad interest to the community